# DSC, FT-IR and NIR with Chemometric Assessment Using PCA and HCA for Estimation of the Chemical Stability of Oral Antidiabetic Drug Linagliptin in the Presence of Pharmaceutical Excipients

**DOI:** 10.3390/molecules27134283

**Published:** 2022-07-03

**Authors:** Anna Gumieniczek, Anna Berecka-Rycerz, Hanna Trębacz, Angelika Barzycka, Edyta Leyk, Marek Wesolowski

**Affiliations:** 1Department of Medicinal Chemistry, Faculty of Pharmacy, Medical University of Lublin, Jaczewskiego 4, 20-090 Lublin, Poland; annabereckarycerz@umlub.pl; 2Department of Biophysics, Faculty of Medicine, Medical University of Lublin, Jaczewskiego 4, 20-090 Lublin, Poland; hanna.trebacz@umlub.pl (H.T.); angelika.barzycka@umlub.pl (A.B.); 3Department of Analytical Chemistry, Faculty of Pharmacy, Medical University of Gdansk, Gen. J. Hallera 107, 80-416 Gdansk, Poland; edyta.leyk@gumed.edu.pl (E.L.); marek.wesolowski@gumed.edu.pl (M.W.)

**Keywords:** linagliptin, excipients, interactions, high temperature/high humidity, DSC, FT-IR, NIR, chemometrics

## Abstract

Pharmaceutical excipients should not interact with active substances, however, in practice, they sometimes do it, affecting the efficacy, stability and safety of drugs. Thus, interactions between active substances and excipients are not desirable. For this reason, two component mixtures of oral antidiabetic drug linagliptin (LINA) with four excipients of different reactivity, i.e., lactose (LAC), mannitol (MAN), magnesium stearate (MGS) and polyvinylpyrrolidone (PVP), were prepared in a solid state. A high temperature and a high humidity of 60 °C and 70% RH, respectively, were applied as stressors in order to accelerate the potential interactions between LINA and excipients. Differential scanning calorimetry (DSC) as well as Fourier transform infrared (FT-IR) and near infrared (NIR) spectroscopy were used to estimate the changes due to potential interactions. In addition, chemometric computation of the data with principal component analysis (PCA) and hierarchical cluster analysis (HCA) was applied to adequately interpret the findings. Of the excipients used in the present experiment, all of them were not inert in relation to LINA. Some of the interactions were shown without any stressing, whereas others were observed under high-temperature/high-humidity conditions. Thus, it could be concluded that selection of appropriate excipients for LINA is very important question to minimize its degradation, especially when new types of formulations with LINA are being developed and manufactured.

## 1. Introduction

Nowadays, gliptins are accepted as a standard oral therapy for type 2 diabetes mellitus. By inhibition of the enzyme known as dipeptidyl peptidase 4 (DPP-4), they prolong insulin secretion in the pancreatic beta cells via glucagon-like peptide 1 and support physiological blood glucose control by insulin [1,2]. All DPP-4 inhibitors are specific for DPP-4 with a high affinity to the enzyme, although their binding to it and pharmacokinetic properties differ to some extent within the group [3]. They can also be divided, according to their chemical structures, into those that mimic the DPP-4 molecule (e.g., vildagliptin and saxagliptin) and those that do not (e.g., sitagliptin, alogliptin and linagliptin). Among the latter, linagliptin (LINA) has a unique chemical structure based on xanthine (Figure 1) and unique kinetics in diabetic patients. In addition, it shows additional protective effects in concomitant vascular problems [4]. For these reasons LINA will be more and more present on the pharmaceutical market in many new pharmaceutical preparations.

As far drug products are concerned, impurities of active pharmaceutical ingredients (APIs) arising from potential interactions with excipients should be taken into account in every new project. For this purpose, some guidelines from existing documents could be used [5,6,7], although many communities, including producers of excipients, postulate the development of unified and more detailed protocols for such research [8]. In typical drug-excipient compatibility experiments, an API is stored in the presence of excipients under stressed conditions, e.g., high temperature and humidity. Finally, many analytical methods such as differential scanning calorimetry, thermogravimetric analysis, powder X-ray diffraction and spectroscopic techniques can be used to assess the drug–excipient compatibility [9]. It is well-known that active substances can react with excipients mainly due to hydrolysis and redox reactions. Excipients can have acidic or basic, as well as oxidative properties, depending on their chemical nature, and may contribute to the instability of pH and redox sensitive drugs. Excipients may also affect drug stability by altering the moisture content, which greatly increases the possibility of drug degradation. In addition, further impurities present in excipients could be the cause of adverse chemical reactions. Thus, many guidelines emphasize the importance of testing the chemical stability of drugs as in their bulk substances, as well as in their final pharmaceutical products with particular respect to excipients [10].

In the literature, a few papers on visible interactions of some gliptins, e.g., sitagliptin and vildagliptin, with pharmaceutical excipients were published [11,12]. However, there is only one paper in this area as far as LINA is concerned in physical mixtures with two polymers, chitosan and gelatin. The optimal formulations were tested for stability at accelerated conditions (40 °C/75% RH) over a period of 1 month and finally examined by Fourier transform infrared (FT-IR) and differential scanning calorimetry (DSC) methods [13].

It is obvious that the experimental data concerning LINA and its interactions with different pharmaceutical excipients is too scarce. What is more, there are no reports in which chemometric methods were used to estimate the potential interactions of this valuable drug with excipients. Thus, the goal of the present study was to examine the chemical stability of LINA in the presence of lactose (LAC), mannitol (MAN), magnesium stearate (MGS) and polyvinylpyrrolidone (PVP). These excipients were chosen based on the literature data and the results of previous experiments concerning gliptins [11]. The solid mixtures of LINA with the above-mentioned excipients were prepared and then stressed with high temperature (60 °C) and high humidity (70% RH), and analyzed using DSC, FT-IR and NIR methods together with chemometric assessment, using principal component analysis (PCA) and hierarchical cluster analysis (HCA).

## 2. Results

At the beginning, differences observed between the stressed and non-stressed pure substances, i.e., LINA as well as LAC, MAN, MGS and PVP, were taken into account in all considerations. In the next step, the differences between the stressed and non-stressed binary mixtures of LINA with the above pharmaceutical excipients were examined. In the respective figures, DSC curves, FT-IR and NIR spectra for all the stressed and non-stressed samples are presented to better show the effects of high-temperature and high-humidity conditions. In addition, DSC, FT-IR and NIR characteristics of pure non-stressed compounds, e.g., LINA, LAC, MAN and PVP are presented in Table 1. Within the corresponding ranges, the presented data are similar to those described in the literature [14].

### 2.1. DSC Results

DSC curves of LINA and excipients are presented in Figure 2, both of the non-stressed samples (upper curves) and of the stressed ones (lower curves). The curve of non-stressed LINA (Figure 2, curve a(I)) reveals two broad endotherms, the first with the peak at 113.0 °C and the second one (much smaller) with the peak at 198.7 °C. Both peaks are asymmetric with broader arms at lower temperatures. After high-temperature/high-humidity stress the endotherms of LINA changed (Figure 2, curve a(II)), revealing their complex nature. The peak in the first endotherm moved to 83 °C and was overlapped with another peak around 110 °C. The shape of the second endotherm suggests two peaks at 196 °C and 202 °C with enthalpy around 20 J/g, which was larger than the same for the non-stressed sample (around 10 J/g). In addition, some interesting observations and comparisons were made using DSC curves of pure excipients. It was noticed that some thermal processes of non-stressed MGS and PVP took place in the same temperature ranges as the first endotherm of pure LINA. When the stressed excipients were examined, it was observed that high-temperature and high-humidity stress did not influence thermal processes of LAC, MAN and MGS. However, the curve of the stressed PVP is to some extent different than that of the non-stressed compound, i.e., the peak in the first endotherm moved to a lower temperature, around 84 °C (Figure 2).

The non-stressed physical mixtures of LINA with each of four excipients (Figure 3, upper curves) revealed some changes concerning LINA, suggesting its potential interactions with the second component at ambient conditions. In the mixtures of LINA with LAC, LINA with MAN, and LINA with PVP, the first LINA endotherm was observed at higher temperatures, i.e., at 114.9 °C, 115.4 °C and 114.7 °C, respectively. It seems that the presence of these excipients slightly increased the thermal stability of LINA. As far as the mixture of LINA and MGS is concerned, the first endotherm of LINA was due to fusion with the complex endothermal events of MGS between 90 °C and 130 °C. All these changes were also observed in respective stressed mixtures (Figure 3, lower curves). Apparent changes were also noticed in the second endotherm of LINA in the mixtures with LAC and with MAN. When the non-stressed mixture of LINA with LAC was examined, the endotherms of LINA (198.7 °C) and LAC (218.2 °C) merged into one event with a peak at 207.5 °C. In the stressed mixture, this fused endotherm deteriorated to lower temperatures with a peak at 202.7 °C. When the non-stressed mixture of LINA with MAN was analyzed, the temperature of the second endotherm for LINA was lowered by about 10 °C to around 188 °C. In the stressed sample, the temperature of this endothermal process of LINA was even lower. It can be concluded that MAN weakened the thermal stability of LINA related to the second endotherm and this effect was strengthened after high-temperature/high-humidity stress.

### 2.2. FT-IR Experiments

FT-IR characteristics obtained for LINA and four excipients, i.e., LAC, MAN, MGS and PVP, are shown in Table 1 and Figure 4 and Figure 5. From the spectra obtained, it was clearly seen that LINA changed slightly after stressing by high temperature/high humidity. Especially, the peaks at 1697 cm^−1^ and 1506 cm^−1^ were affected (Figure 4).

As far as excipients are concerned, LAC, MAN and MGS did not show changes under stress conditions. However, slight changes were observed in the spectrum of PVP, especially in the region 1600–1500 cm^−1^ (Figure 5).

As far as the binary mixtures of LINA with excipients are concerned, some interactions were observed without stressing, whereas other changes occurred only under high-temperature/high-humidity conditions (Figure 6). The stressed mixture of LINA and LAC showed potential interactions such as decreasing of characteristic peaks of LINA at 3331 cm^−1^ and 3285 cm^−1^ due to primary amine group (N–H stretching). In addition, broadening the peaks at 2944 cm^−1^ (C–H stretching), 1654 cm^−1^ (C=O stretching) and 1506 cm^−1^ (C=C stretching), as well as changing the shape of the peaks of LINA in the region 1450–1100 cm^−1^ (C=C stretching and C–N stretching vibrations) was observed. Bearing in mind the above results, especially the broadening the peak at 1654 cm^−1^, a new vibration due to imine formation (1690–1640 cm^−1^) could be supposed and the possibility of chemical interaction via Maillard’s reaction between the amine group of LINA and LAC could be suggested. In the literature, similar interactions with LAC were reported for other drugs from gliptins, i.e., for sitagliptin containing a primary amine group and vildagliptin containing a secondary amine group [11,12].

When the non-stressed mixture of LINA and MAN was studied, the peak of LINA at 1506 cm^−1^ visibly changed its shape. In addition, the peaks of LINA at 1436 cm^−1^ and MAN at 1450 cm^−1^ were observed as overlapped. After high-temperature/high-humidity stressing, these overlapped peaks changed their shapes and deteriorated to lower wavenumbers, suggesting a new sort of interaction. However, the peaks due to the –OH groups of MAN did not change visibly. Thus, the possibility of hydrogen bond formation between the amine group of LINA and –OH groups of MAN cannot be suggested, contrary to our previous results obtained for sitagliptin [11] (Figure 6).

The spectrum of the non-stressed mixture of LINA and MGS showed a lack of peaks at 3331 cm^−1^, 3285 cm^−1^ and 2944 cm^−1^ due to N–H stretching and C–H stretching of LINA. When the mixture was treated with high temperature/high humidity, one more change occurred, i.e., the lack of the peak of LINA of at 1506 cm^−1^. Thus, the interactions via hydrogen bonding between the amine group of LINA and the carboxylic group of MGS could be suggested. What is more, the characteristic peak of MGS at 1572 cm^−1^ (–COOH group) was also changed in the stressed mixture (Figure 6).

As far as the mixture of LINA with PVP is concerned, the spectrum of the non-stressed mixture showed a lack of peaks at 3331 cm^−1^, 3285 cm^−1^ and 2944 cm^−1^ due to N–H stretching and C–H stretching of LINA, and overlapping of the peaks of LINA at 1697 cm^−1^ and PVP at 1652 cm^−1^. When the mixture treated with high temperature/high humidity was examined, these overlapped peaks were significantly broadened up to 1613 cm^−1^. In addition, the spectrum of the stressed mixture shows changes at 1291 cm^−1^ and 1287 cm^−1^ (the peaks of LINA due to C–N stretching) and at 1291 cm^−1^ (the peak of PVP due to C–O stretching) (Figure 6). Thus, it could be suggested that the NH, CN and CO groups of LINA and the CO groups of PVP could be involved in some interactions. Previously, the changes concerning the peak due to secondary amine of vildagliptin due to interactions with PVP were observed [12]. What is more, the possibility of hydrogen bonding via the oxygen atom in PVP for many drug molecules was described in the literature [15].

### 2.3. NIR Experiments

NIR characteristics obtained for non-stressed LINA and four excipients are shown in Table 1. From the obtained NIR spectra it is seen that LINA did not change under high-temperature/high-humidity stress. Also, the NIR spectra of excipients did not show visible changes after stressing (Figure 7 and Figure 8).

When the NIR spectrum of the binary mixture of LINA with LAC was examined, it was observed that there were no significant changes in the nature of the absorption patterns under ambient conditions. On the contrary, in the binary mixture of LINA and MAN, overlapping of the peak of LINA at 4925 cm^−1^ with the bands of MAN is clearly seen. When the non-stressed mixture of LINA and MGS was analyzed, the spectrum showed overlapping the bands of LINA at 5753 cm^−1^ and 5651 cm^−1^ with the bands of MGS at 5776 cm^−1^ and 5665 cm^−1^. Also, the peaks in the region 4450–4300 cm^−1^ changed their shapes and overlapped. As far as the mixture of LINA and PVP is concerned, it could be observed that the characteristic peaks of LINA at 5193 cm^−1^ and of PVP at 5175 cm^−1^ are overlapped. In addition, the peak of LINA at 4925 cm^−1^ is visibly decreased (Figure 9).

What is more, all of the examined mixtures showed potential interactions among the components after stressing. When the stressed mixture of LINA and LAC was considered, the main changes observed were the disappearance of the characteristic peak of LINA at 4925 cm^−1^ and the changing of the peaks in the region 4400–4300 cm^−1^. When the stressed mixture of LIN and MAN was analyzed, the additive change was observed as the lack of three characteristic peaks of LINA around 5190 cm^−1^. When the stressed mixture of LINA and MGS was taken into account, the main change seen was the disappearance of the peak of LINA at 4925 cm^−1^. As far as the stressed mixture of LINA and PVP is concerned, almost all changes that were seen without any stressing, were intensified under high-temperature and high-humidity stress (Figure 9).

### 2.4. Multivariate Statistical Calculations

To improve the interpretation of DSC curves, as well as FT-IR and NIR spectra, two unsupervised multivariate statistical methods were used, principal component analysis (PCA) and hierarchical cluster analysis (HCA) [16]. PCA is based on the dimensionality reduction of huge data sets. This increases the interpretability of data under study with minimal loss of information. On the other hand, HCA shows the tendency of samples to form clusters, that is to create groups of samples with similar characteristics. The findings obtained during multivariate assessment of thermal (DSC) and spectroscopic (FT-IR, NIR) data for LINA, excipients and their binary physical mixtures, stored under ambient and stressed conditions, are summarized in Table 2.

In general, these data reveal that interpretation of DSC and FT-IR data leads to similar conclusions, regardless of the fact that DSC curves reflect such phase transitions that occurred in substances when heating (the characteristic temperatures and heats of transitions), whereas FT-IR spectra reflect the chemical structure of substances under study (the chemical bonds and functional groups).

Findings obtained using NIR spectra differ slightly from those obtained using DSC and FT-IR measurements. Probably, this is due to the fact that NIR spectra are based on molecular overtone and combination vibrations, hence, the absorption bands are typically quite small and very broad. This leads to complex spectra, in which it is difficult to attribute a specific band to a specific chemical compound.

In general, PCA calculations for the data acquired from the DSC curves revealed that stressed conditions have a higher impact on the stability of binary mixtures than that on the stability of LINA and pharmaceutical excipients alone. However, some differences occurred among respective compounds and mixtures. As shown in Figure 10 (PCA), the non-stressed and stressed samples of MAN, LAC and MGS are grouped into three separate clusters I, II and III, respectively, at very similar values of the first two principal components (PC1 and PC2). This implies that high temperature and high humidity did not affect the thermal stability of these excipients. In contrast to this, the stressed conditions strongly affected the thermal stability of pure LINA and PVP. Both compounds and their stressed counterparts are localized in cluster IV, at very different PC1 and PC2 values. Such localization of these samples confirms that DSC curves of the stressed and non-stressed samples are not the same. They display physical or chemical changes that occurred in a sample under the influence of high temperature and high humidity. Additionally, Figure 10 (PCA) shows that, besides excipients, all binary mixtures of LINA, i.e., with MAN, LAC, MGS and PVP (non-stressed and stressed as well), are also placed in clusters I, II, III and IV, respectively. However, excluding the mixture of LINA with PVP (cluster IV), the remaining samples are localized at slightly different values of PC1 or PC2 as compared with the excipients alone. Similarly, as in the case of PVP alone, high temperature and high humidity affect, to the greatest extent, the stability of the mixture of LINA with PVP. Thus, such organization of the PCA clusters proves that excluding PVP, the rest of excipients have a slight effect on the thermal stability of LINA, also taking into account the stressed conditions.

To avoid misinterpretation of the thermal data, hierarchical cluster analysis (HCA) was also used for the analysis of the DSC curves as a second multivariate statistical method. In general, the findings of the HCA calculations are consistent with those obtained using PCA. Figure 10 (HCA) shows that non-stressed and stressed samples of LINA, excipients alone (MAN, LAC, MGS and PVP), and LINA binary mixtures with these excipients are linked together in the same way, as was the case for the PCA score scatter plot (Figure 10 (PCA)). MAN and LAC, and their mixtures with LINA, form clusters I and II, respectively. Cluster III created on the HCA tree diagram could be divided into two subclusters, IIIA and IIIB. The former cluster contains the non-stressed and stressed MGS and its mixture with LINA, whereas the latter is created by the non-stressed and stressed LINA and PVP alone, and their mixture. Thus, the HCA diagram reveals that the four excipients and their mixtures with LINA create four clusters (I-IV), taking into account similarities in their thermal properties.

The findings of the PCA calculations for the FT-IR data are quite similar to those obtained for the DSC data. High temperature and high humidity affected the thermal stability of mixtures to a greater extent than the stability of LINA and excipients alone. Figure 11 (PCA) shows that similarly, as in the case of DSC data the non-stressed and stressed samples of LINA, PVP and the mixture of LINA with PVP are grouped together in a separate cluster, I. Distribution of LINA and PVP at different values of PC1 and PC2 implies that high temperature and high humidity affected their stability. Furthermore, the non-stressed and stressed samples of MGS and its mixture with LINA create cluster IV. Scattering of the samples within this cluster may suggest the impact of stressed conditions on the stability of this mixture. Particular attention should also be paid to the distribution of the LAC and MAN samples, and their mixtures with LINA. The stressed and non-stressed samples of LAC and MAN alone create a separate cluster III, whereas their mixtures with LINA are grouped in cluster II. This could be due to the fact that MAN is a sugar alcohol with six hydroxy groups ((2R,3R,4R,5R)-hexane-1,2,3,4,5,6-hexol), whereas LAC is a sugar composed of galactose and glucose (β-D-galactopyranosyl-(1→4)-D-glucose) that has eight hydroxy groups per molecule. As the dominant absorption bands are assigned to hydroxy groups, both excipients have similar FT-IR spectra. Thus, they show a tendency to create a cluster of samples with similar characteristics. The scattering of samples within the cluster II implies that high temperature and high humidity strongly affected thermal stability, especially stability of the mixture of LINA with LAC. As mentioned in the section “FT-IR Experiments”, this may be due to chemical interaction via Maillard’s reaction between the amine group of LINA and LAC.

Overall, the FT-IR spectra revealed that stressed conditions may induce physical or chemical changes in LINA and PVP alone, and potential interactions in the mixtures of LINA with all excipients examined. i.e., LAC, MAN, MGS and PVP.

As in the case of DSC data, the findings for HCA calculations based on the FT-IR data are, in general, consistent with those obtained by PCA. As shown in Figure 11 (HCA), the non-stressed and stressed samples of LINA, excipients alone (MAN, LAC, MGS and PVP), and LINA binary mixtures with excipients are grouped together in separate clusters on the lowest linkage levels, below 10% on the similarity axis. At higher linkage levels, the samples of MGS and PVP, and their mixtures with LINA create separate clusters, I and II, respectively. Furthermore, samples of MAN and LAC are grouped in a separate cluster, IV, whereas their mixtures with LINA are localized in the cluster IIIB. Hence, HCA calculations using FT-IR data may imply that stressed conditions could induce physical or chemical changes in LINA and PVP alone, and potential interactions in the mixtures of LINA with LAC, MAN and PVP.

The PCA score scatter plot for the data acquired from NIR spectra (Figure 12 (PCA)) differs greatly from the PCA plots developed for the DSC and FT-IR data (Figure 10 (PCA) and Figure 11 (PCA)). As shown in Figure 12 (PCA), there are eight small clusters, seven of them are localized at a narrow range of the PC1 values, i.e., between −0.85 and −1.0. Each of the clusters I, II, III, IV and VII comprise only two samples, that is the non-stressed and stressed MAN, LAC, LINA mixture with MAN, LINA mixture with LAC, and MGS, respectively. The values of PC1 or PC2 suggest that stressed conditions affect merely the stability of the LINA mixture with LAC, and MGS alone. As in the case of PCA plots developed for DSC and FT-IR data (Figure 10 (PCA), cluster IV; Figure 11 (PCA), cluster I), two neighboring clusters consisted of stressed and non-stressed samples of LINA and the LINA mixture with PVP (cluster V), and PVP alone (cluster VI). In the last cluster, the non-stressed LINA mixture with MGS is also placed, however, the reason for localization of this mixture in cluster VI is difficult to explain. On the other hand, the stressed LINA mixture with MGS is localized on the opposite side of the PCA score scatter plot (cluster VIII). This implies that high temperature and high humidity had a very strong influence on the stability of this mixture.

The HCA tree diagram developed for the NIR data is more complicated (Figure 12 (HCA)) than the diagrams based on the DSC and FT-IR data (Figure 10 (HCA) and Figure 11 (HCA)). The mixture of LINA with PVP (cluster I) and the mixture of LINA with MGS, both stressed, and the mixture of LINA with PVP (cluster IIA) differ significantly from each other and from the remaining mixtures (Figure 12 (HCA)). This suggests a strong effect of high temperature and high humidity on their stabilities. Moreover, the stressed mixtures of LINA with MAN and LINA with LAC grouped in cluster IIB differ completely from their counterparts localized in cluster IIIB. This also suggests a strong effect of stressed conditions on their stability. Unlike these samples, localization of the non-stressed and stressed LINA and excipients (LAC, MAN, MGS, PVP) in two-components subclusters (IIB, IIIA, and IIIB) created on the lowest linkage levels, below 8% on the distance axis, indicates that the stressed conditions did not have an impact on the stability on these compounds.

Concluding, the PCA score scatter plots and the HCA tree diagrams developed for the thermal and spectroscopic data allowed us to conclude that excipients, especially under stressed conditions, affect the stability of LINA in the mixtures.

Furthermore, to identify which absorption bands of the FT-IR spectra of LINA underwent significant changes due to mixing with excipients (LAC, MAN, MGS and PVP), and after high-temperature/high-humidity stressing, additional PCA calculations were performed using four matrices (A, B, C and D) described in detail in the section “Chemometrics”. This approach was previously used in the study of methylxanthine mixtures with hydroxypropyl methylcellulose [17]. The findings of the PCA calculations for LINA and its mixture with excipients are graphically presented in Figure 13 and the data obtained are compiled in Table 3.

Overall, outcomes of these studies can be obtained by a comparison of the plots developed for PC1 and PC2 values, which were calculated for four successive matrices. As an example, Figure 13 shows the results of PCA calculations using FT-IR data for LINA and its mixture with LAC. Comparing PC1 and PC2 lines calculated using the first matrix (Figure 13, lines A, LINA and LAC alone) with those lines for PC1 and PC2 obtained for the second matrix (Figure 13, lines B, LINA and its mixture with LAC), both under non-stressed conditions, the impact of the excipient (LAC) on the chemical stability of LINA can by assessed by taking into account the changes in absorption bands of LINA. Findings obtained after this comparison for all the excipients used in this study (LAC, MAN, MGS and PVP) are listed in Table 3. On the other hand, comparing PC1 and PC2 lines calculated using the first matrix (Figure 13, lines A, LINA and LAC alone under non-stressed conditions) with those lines for PC1 and PC2 obtained for the third matrix (Figure 13, lines C, LINA and LAC alone under stressed conditions), the effect of high temperature and high humidity on the chemical stability of LINA can be evaluated (Table 3). Additionally, detailed analysis of the PC1 and PC2 lines calculated using the fourth matrix (Figure 13, lines D, LINA and its mixture with LAC under stressed conditions) with PC1 and PC2 lines obtained for each other matrix enable us to determine if there are additional differences that were not seen separately to the effects of the excipient (LAC) or stressed conditions on the drug substance (LINA).

## 3. Materials and Methods

### 3.1. Chemicals

Linagliptin (LINA) from Gute Chemie GmbH (Karlsruhe, Germany), lactose (LAC), mannitol (MAN), magnesium stearate (MGS), and polyvinylpyrrolidone (PVP) from Sigma-Aldrich (St. Louis, MO, USA) were used. The drug and excipients were of pharmaceutical purity and were used as obtained.

### 3.2. Preparation of Samples for Stability Studies

Binary mixtures of LINA with four excipients, i.e., LAC, MAN, MGS and PVP, were prepared by mixing the components in an agate mortar at 1:1 ratio (*w*/*w*). Then, pure LINA, pure excipients and the prepared mixtures were dispersed as uniform 20 mg portions (in duplicate) to standardized small flat vessels (the thickness of the layers was approximately 3 mm). Half of them were placed in a climate chamber KBF-LQC (Binder GmbH, Tuttlingen, Germany) at 60 °C and 70% RH for 60 days, while the rest of the samples (non-stressed) were stored in a desiccator at ambient conditions (23 ± 2 °C). After finishing the accelerated degradation, the stressed samples were transferred from the climate chamber to a desiccator.

### 3.3. Differential Scanning Calorimetry (DSC)

The DSC method was applied to study thermal transformations of pure LINA, pure excipients and all the binary mixtures using a Q200 calorimeter (TA Instruments, New Castle, DE, USA). All the samples, as non-stressed and stressed as well, in similar amounts of 4–5 mg, were closed in aluminum pans (Tzero Pan, TA Instruments) and the temperature was scanned from 20 °C to 250 °C at a heating rate of 10 °C/min in a nitrogen atmosphere. An empty pan was used as a reference. All the measurements were performed in triplicate. The DSC curves were analyzed using TA Universal Analysis software.

### 3.4. FT-IR and NIR Measurements

FT-IR and NIR spectra were recorded on a Nicolet 6700 spectrometer (ThermoScientific, Waltham, MA, USA) equipped with a Smart iTR™ ATR Sampling Accessory and Near IR Integrating Sphere. After recording background spectra, the samples were analyzed over the ranges 4.000–650 cm^−1^ for FT-IR and 10.000–4000 cm^−1^ for NIR measurements. Each spectrum was recorded as an average of four scans. For analyzing all the spectra, OMNIC software from ThermoScientific was applied.

### 3.5. Chemometrics

Two multivariate statistical approaches, principal components analysis (PCA) and hierarchical cluster analysis (HCA) were used as supporting tools for interpretation of the DSC, FT-IR and NIR data. Calculations were performed using Statistica 13.3 software (TIBCO Software Inc., Palo Alto, CA, USA). Matrices for calculations were constructed using thermal and spectral data for LINA, excipients and their binary physical mixtures stored under non-stressed and stressed conditions. The matrix of the DSC data consisted of the heat values measured at every 0.17 °C in the range of 50–225 °C. In the case of FT-IR and NIR measurements, the matrices were created based on the absorbance values acquired at every 0.5 cm^−1^ in the range of 4000–650 cm^−1^, and every 4 cm^−1^ in the range of 10,000–4000 cm^−1^, respectively.

The correlation matrix was used for PCA calculations. The values of the first three principal components (PC1, PC2 and PC3) are listed in Table 4. As the first two PCs explained the highest percentage of total variance, sample classifications were performed using a 2D score scatter plot (PC1 and PC2), without rotation. Ward’s method and Euclidean distance were applied in the case of HCA calculations and findings obtained were visualized as a tree diagram referring to 100% of dissimilarity.

Moreover, for each excipient, i.e., LAC, MAN, MGS and PVP, four matrices were additionally constructed using the data obtained from the FT-IR spectra. The absorbance values acquired at each wavelength were used as columns. The rows of the first matrix (A) contained four spectra of LINA and four spectra of each excipient, whereas the second one (B) included four spectra of LINA and four spectra of LINA mixture with a given excipient. The third (C) and fourth (D) matrices had the same rows as the first and second matrices, but the samples were stored under stressed conditions.

## 4. Conclusions

Four different methods, i.e., DSC, FT-IR, NIR and chemometrics, i.e., PCA and HCA were used to examine potential interactions of oral antidiabetic drug LINA with four excipients in a solid state (Appendix A). It was found that some interactions between components occurred without any stressing, whereas others were shown only under stressing by high temperature and high humidity. Between the excipients used in the present experiment, LAC, MAN and PVP were shown as peculiarly reactive with LINA. Therefore, we can conclude that selection of appropriate excipients for LINA is still important in minimizing their degradation process. This question could be critical when new formulations are being developed. As described above, most noticed interactions may be due to the presence of an amino group in the structure of LINA. This may also be an important question in similar studies on other drugs containing an amine group.

## Figures and Tables

**Figure 1 molecules-27-04283-f001:**
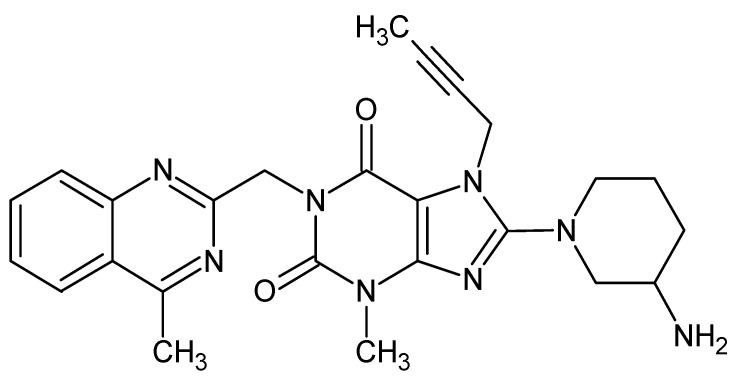
Chemical structure of linagliptin (LINA).

**Figure 2 molecules-27-04283-f002:**
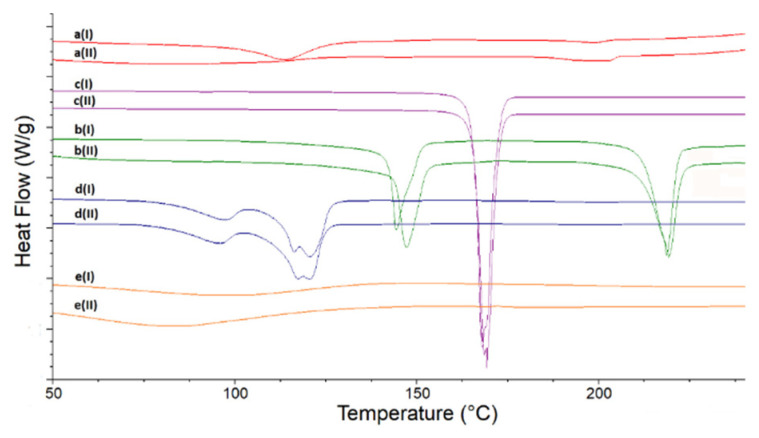
DSC curves for linagliptin (a), lactose (b), mannitol (c), magnesium stearate (d) and polyvinylpyrrolidone (e) in the non-stressed (I) and stressed (II) samples.

**Figure 3 molecules-27-04283-f003:**
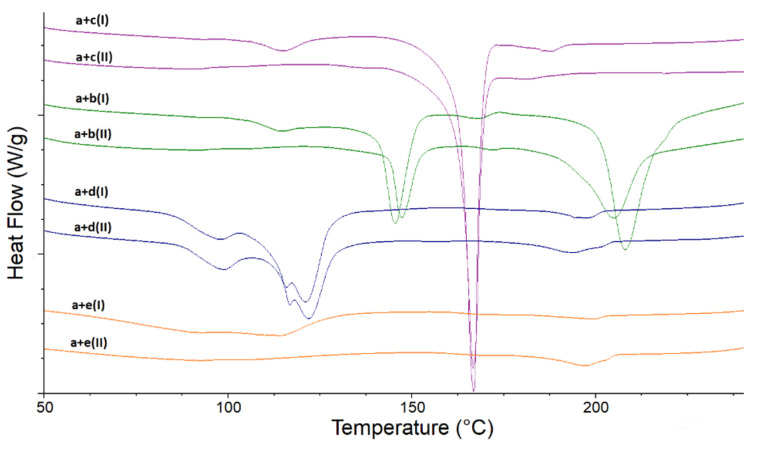
DSC curves for binary mixtures of linagliptin (a) with lactose (b), mannitol (c), magnesium stearate (d) and polyvinylpyrrolidone (e) in the non-stressed (I) and stressed (II) samples.

**Figure 4 molecules-27-04283-f004:**
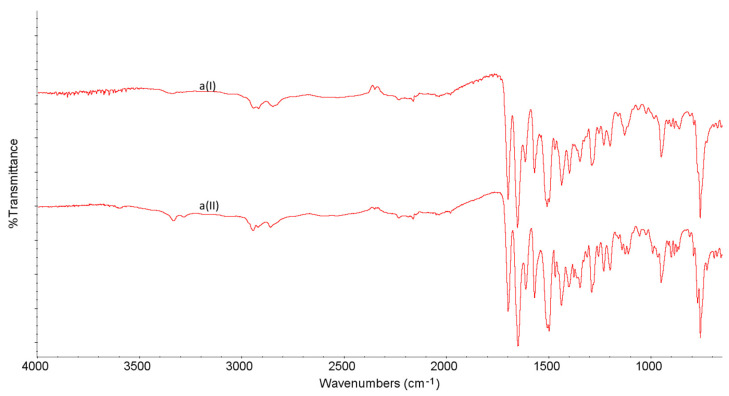
FT-IR spectra for linagliptin (a) in the non-stressed (I) and stressed (II) samples.

**Figure 5 molecules-27-04283-f005:**
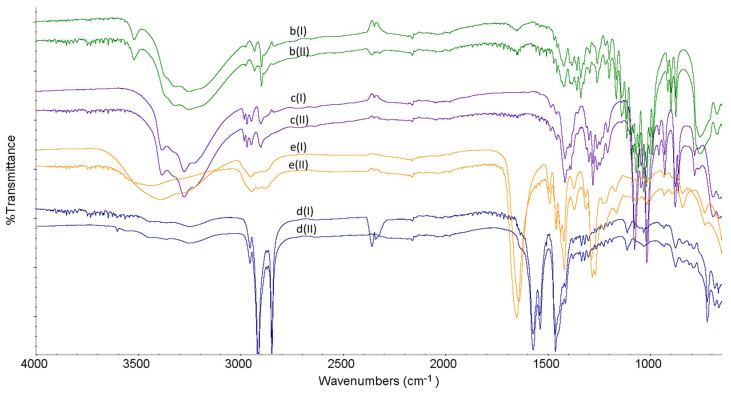
FT-IR spectra for lactose (b), mannitol (c), magnesium stearate (d) and polyvinylpyrrolidone (e) in the non-stressed (I) and stressed (II) samples.

**Figure 6 molecules-27-04283-f006:**
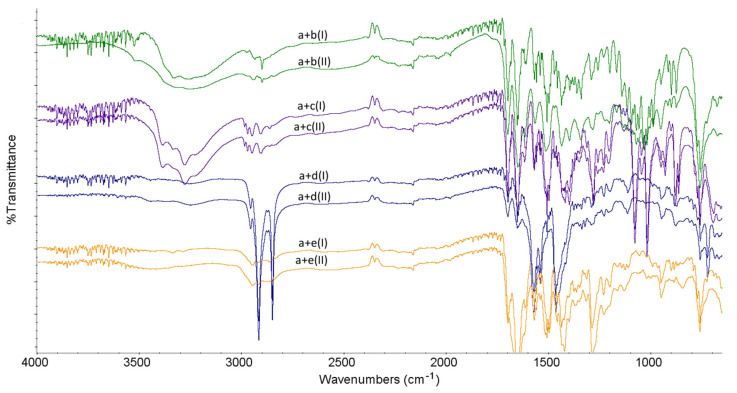
FT-IR spectra for binary mixtures of linagliptin (a) with lactose (b), mannitol (c), magnesium stearate (d) and polyvinylpyrrolidone (e) in the non-stressed (I) and stressed (II) samples.

**Figure 7 molecules-27-04283-f007:**
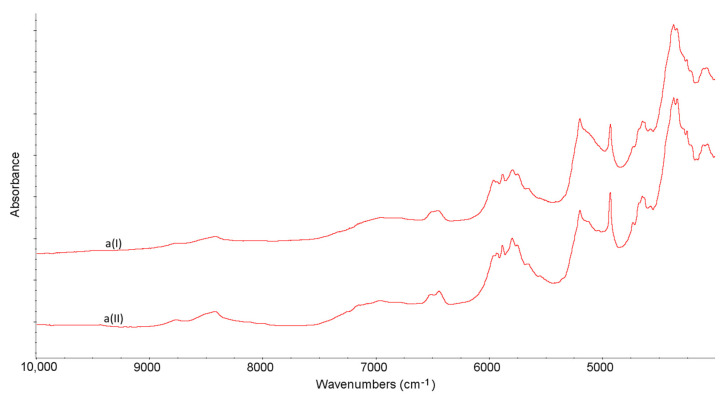
NIR spectra for linagliptin (a) in the non-stressed (I) and stressed (II) samples.

**Figure 8 molecules-27-04283-f008:**
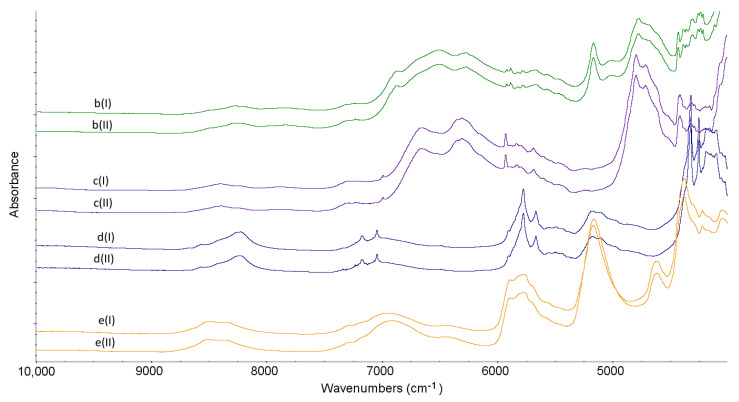
NIR spectra for lactose (b), mannitol (c), magnesium stearate (d) and polyvinylpyrrolidone (e) in the non-stressed (I) and stressed (II) samples.

**Figure 9 molecules-27-04283-f009:**
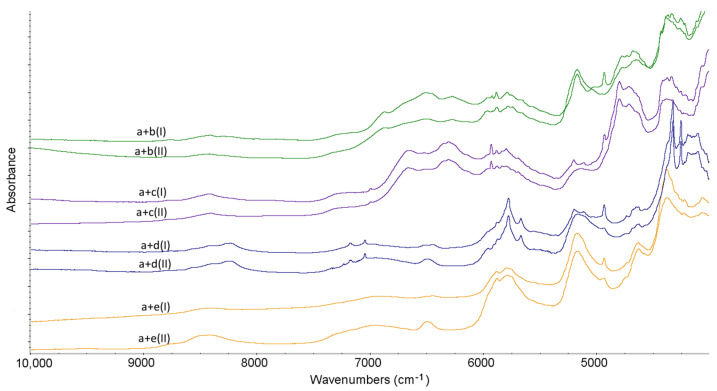
NIR spectra for binary mixtures of linagliptin (a) with lactose (b), mannitol (c), magnesium stearate (d) and polyvinylpyrrolidone (e) in the non-stressed (I) and stressed (II) samples.

**Figure 10 molecules-27-04283-f010:**
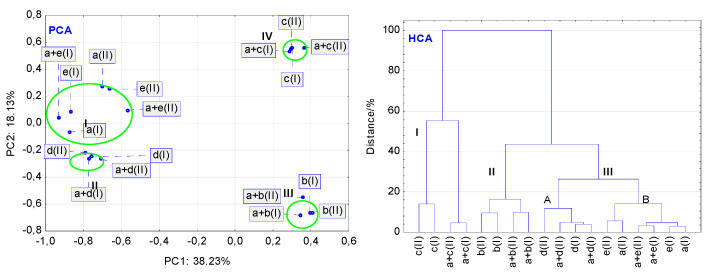
Findings of multivariate calculations using DSC data: PCA score scatter plot, and HCA tree diagram. Linagliptin = a(I) and a(II), lactose = b(I) and b(II), mannitol = c(I) and c(II), magnesium stearate = d(I) and d(II), and polyvinylpyrrolidone = e(I) and e(II). Physical mixtures of linagliptin with lactose = a + b(I) and a + b(II), mannitol = a + c(I) and a + c(II), magnesium stearate = a + d(I) and a + d(II), and polyvinylpyrrolidone = a + e(I) and a + e(II). (I and II) denote the non-stressed and stressed samples, respectively.

**Figure 11 molecules-27-04283-f011:**
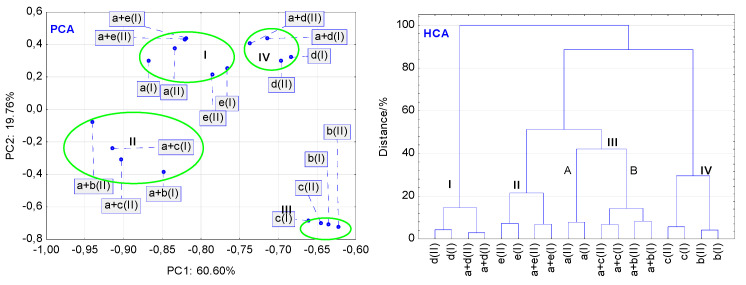
Findings of multivariate calculations using FT-IR data: PCA score scatter plot, and HCA tree diagram. Linagliptin = a(I) and a(II), lactose = b(I) and b(II), mannitol = c(I) and c(II), magnesium stearate = d(I) and d(II), and polyvinylpyrrolidone = e(I) and e(II). Physical mixtures of linagliptin with lactose = a + b(I) and a + b(II), mannitol = a + c(I) and a + c(II), magnesium stearate = a + d(I) and a + d(II), and polyvinylpyrrolidone = a + e(I) and a + e(II). (I and II) denote the non-stressed and stressed samples, respectively.

**Figure 12 molecules-27-04283-f012:**
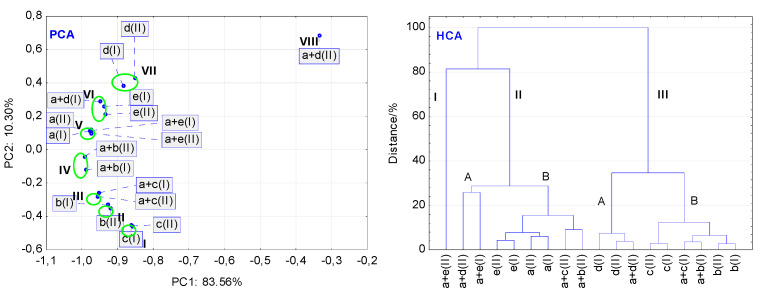
Findings of multivariate calculations using NIR data: PCA score scatter plot, and HCA tree diagram. Linagliptin = a(I) and a(II), lactose = b(I) and b(II), mannitol = c(I) and c(II), magnesium stearate = d(I) and d(II), and polyvinylpyrrolidone = e(I) and e(II). Physical mixtures of linagliptin with lactose = a + b(I) and a + b(II), mannitol = a + c(I) and a + c(II), magnesium stearate = a + d(I) and a + d(II), and polyvinylpyrrolidone = a + e(I) and a + e(II). (I and II) denote the non-stressed and stressed samples, respectively.

**Figure 13 molecules-27-04283-f013:**
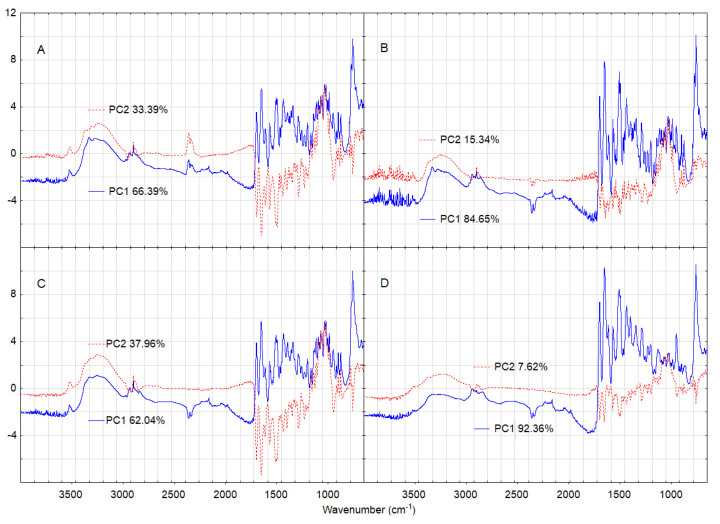
The first and second principal components (PC1 and PC2) calculated using matrices containing the FT-IR data set for: (**A**) linagliptin and lactose under non-stressed conditions, (**B**) linagliptin and its mixture with lactose under non-stressed conditions, (**C**) linagliptin and lactose under stressed conditions, and (**D**) linagliptin and its mixture with lactose under stressed conditions.

**Table 1 molecules-27-04283-t001:** DSC, FT-IR and NIR characteristics of pure compounds: linagliptin (LINA), lactose (LAC), mannitol (MAN), magnesium stearate (MGS) and polyvinylpyrrolidone (PVP).

Compound		DSC *		FT-IR		NIR
		T (°C)	Enthalpy (J/g)	(cm^−1^)	Vibrations	(cm^−1^)
LINA a(I)	Peak 1	113.0 ± 1.0	163.4 ± 5.1	3331	N–H stretching	5938
				3285	N–H stretching	5929
				2944	C–H stretching	5753
	Peak 2	198.7 ± 0.5	10.8 ± 0.8	1697	C=O stretching	5651
				1654	C=O stretching	5193
				1564	N–H bending	4925
				1450		4642
				1291	C–N stretching	4369
				1287	C–N stretching	4254
LAC b(I)	Peak 1	145.4 ± 2.0	166.4 ± 2.1	3527	O–H stretching	6873
				3325		6507
				2897	C–H stretching	5915
	Peak 2	218.2 ± 0.5	150.0 ±3.6	1419		5776
				1337		5165
				1202		5026
				1143		4772
				1091		4434
				1000	C–O stretching	4305
MAN c(I)		167.7 ± 0.6	302.2 ± 6.0	3382		6664
				3200	O–H stretching	5924
				2969		5836
				1450	C–H bending	5684
				1417		4795
				1291	C–H bending	4712
				1277		4416
				1076	C–O stretching	
				1016	C–O stretching	
MGS d(I)	Peak 1	98.4 ± 1.0	255.0 ± 2.8 for three peaks together	3253		8219
				3087	C–H stretching	7169
	Peak 2	115.9 ± 0.2		2956	C–H stretching	7044
				2915	C–H stretching	5776
	Peak 3	121.3 ± 0.6		2848		5651
				1572		5489
				1543	C=O stretching	5179
				1450		5096
PVP e(I)		98.2 ± 2.2	186.6 ± 10.7	3442		8506
				2969	C–H stretching	8344
				1652	C=O stretching	5892
				1491		5758
				1455	C–H bending	5175
				1373	C–C stretching	4619
				1291	C–O stretching	4226
				1280	C–N bending	

* Mean ± standard deviation (SD). Roman numeral I denotes the non-stressed samples.

**Table 2 molecules-27-04283-t002:** Effects of high temperature/high humidity (60 °C/70% RH) on linagliptin (LINA), excipients (LAC, MAN, MGS and PVP) and their binary mixtures.

Sample	DSC		FT-IR		NIR	
	PCA	HCA	PCA	HCA	PCA	HCA
LINA, a(II)	+++	++	++	+	–	–
LAC, b(II)	–	–	–	–	–	–
MAN, c(II)	–	–	–	–	–	–
MGS, d(II)	–	++	–	–	++	–
PVP, e(II)	+++	++	+	+	+	–
LINA + LAC, a + b(II)	+	+	+++	+	++	+++
LINA + MAN, a + c(II)	+	++	+	+	–	+++
LINA + MGS, a + d(II)	+	–	+	–	+++	+++
LINA + PVP, a + e(II)	+++	–	–	+	–	+++

The influence of the stressed conditions: +++ strong, ++ medium, + week, – no impact. Roman numeral II denotes the stressed samples.

**Table 3 molecules-27-04283-t003:** Effect of excipients and stress conditions on linagliptin based on the PC1 and PC2 values calculated using FT-IR data.

Characteristic Spectral Vibrations of LINA (cm^−1^)	LINA + LAC	LINA + MAN	LINA + MGS	LINA + PVP
N–H stretching at 3331	*	–	*	*
N–H stretching at 3285	*	–	*	*
C–H stretching at 2944	*	–	–	–
~2300	+	–	+	–
C=O stretching at 1697	++	++	–	++
C=O stretching at 1654	++	++	–	++
N–H bending at 1564	++	–	–	++
1506	++	–	–	++
1450	–	++	–	–
1436	–	++	–	–
C–N stretching at 1291	–	–	–	–
C–N stretching at 1287	–	+	–	–
~950	*	–	–	–
~900	–	*	–	–
~750	–	–	+	–

The influence of excipients: ++ strong, + week, – no impact. The influence of stressed conditions: * strong, – no impact.

**Table 4 molecules-27-04283-t004:** Values of the first three principal components (PC1, PC2 and PC3) calculated for thermal and spectral data sets.

Method	Variance (%)		
	PC1	PC2	PC3
DSC	38.23	18.13	16.42
FT-IR	60.60	19.76	11.77
NIR	83.56	10.30	3.68

## Data Availability

Not applicable.

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
