# Peer review of "DSC, FT-IR and NIR with Chemometric Assessment Using PCA and HCA for Estimation of the Chemical Stability of Oral Antidiabetic Drug Linagliptin in the Presence of Pharmaceutical Excipients"

_molecules, 2022, doi:10.3390/molecules27134283_

Round 1

Reviewer 1 Report

The article “DSC, FT-IR and NIR with chemometric assessment using PCA and HCA for estimation of chemical stability of oral antidiabetic drug linagliptin in the presence of pharmaceutical excipients” by Gumieniczek et al. describes a study of the effects of different compositions and environmental conditions on the stability of the drug molecule linagliptin. The first part of the manuscript reports on experimental results, the second part uses multivariate statistical calculations (PCA and HCA) to analyze the results.

The paper is well written, and the experimental methodology seems to be in accordance with other studies of this kind. The list of references is short, relevant references to drug regulating agencies (FDA, EMA, CDSCO etc.) and their testing requirements are missing. The experimental data supports the conclusions of the manuscript while the statistical analysis seems not always to be in line with the observed effects.

1.       The presentation of the experimental data could be improved by plotting the subtracted data (stressed minus non-stressed datasets, especially for the IR data) in addition to the presented data. This would allow for an easy analysis of the IR bands that get affected by mixing/heating.

2.       The FTIR data displays a change of background CO2 levels (band around 2340 cm-1) due to insufficient purging of the FTIR instrument. The corresponding region should not be used for the multivariate statistical calculations.

3.       The data for a+b(II) in Figure 6 seems to have been recorded with lower resolution than the other traces.

4.       What are the criteria (value range) to qualify the stressed condition as “strong, medium, weak or no impact”?

5.       Please add the lettering scheme for the compounds (a(I) through a+e(II) to the column “Compound” in Table 1 and “sample” in Table 2. This allows for a faster comparison of the table entries to the Figures.

6.       Figure 13 would benefit from a slight overhaul, e.g. axis labels in increments of e.g. 500 cm-1. Please mention FTIR data in the figure caption. Please show the same data ranges for Figure 13 and Figure 6.

7.       Fig. 13 A and C, and B and D look very similar while Table 2 lists the changes between stressed and unstressed as strongest for this mixture. Please explain how this analysis translates into the “strong, medium, weak or no impact” categorization.

8.       The results for LINA+LAC and LINA+PVP in Table 2 and Table 3 seem to contradict each other. Both show similar behavior in their PC1 and PC2 values, but Table 2 indicates strong vs. no impact in the IR.

Author Response

The paper is well written, and the experimental methodology seems to be in accordance with other studies of this kind. The list of references is short, relevant references to drug regulating agencies (FDA, EMA, CDSCO etc.) and their testing requirements are missing. The experimental data supports the conclusions of the manuscript while the statistical analysis seems not always to be in line with the observed effects.

Thank you for these comments. Previously, our list of references was based on similar experimental works in the area of stability of gliptins. In the current version, we added a few sentences and respective 5 literature items. They are related to official guidelines concerning drug stability (ICH guidelines) or contain scientific reviews in this area.

As far as statistical analysis is concerned, some changes were introduced in our revised text.

  1. The presentation of the experimental data could be improved by plotting the subtracted data (stressed minus non-stressed datasets, especially for the IR data) in addition to the presented data. This would allow for an easy analysis of the IR bands that get affected by mixing/heating.

Thank you for your suggestion. We agree that it could be useful to use the spectral subtraction function to show the difference between the stressed and non-stressed mixtures. However, since it is not used routinely in works on potential drug-excipient interactions, we did not plan it for our experiment. The introduction of such spectra processing at the present stage would require the repetition of all our calculations concerning FT-IR and fundamental changes in the presented publication.

  1. The FTIR data displays a change of background CO2 levels (band around 2340 cm-1) due to insufficient purging of the FTIR instrument. The corresponding region should not be used for the multivariate statistical calculations.

Our experiments did not include any mathematical corrections of the obtained FT-IR or NIR spectra. At the same time, we ensure that all spectra were measured in similar conditions and the background measurement was performed after each hour of work.

We did not make any corrections to prevent a possible loss of information about the compared spectra. Therefore, we decided to include the entire measured ranges in our chemometric analysis.

  1. The data for a+b(II) in Figure 6 seems to have been recorded with lower resolution than the other traces.

Please be assured that all our spectra were measured under similar conditions with the same resolution. Actually, visible changes were observed when we compared the mixture” a+b” before and after stressing. Therefore, in the Figure 6, this spectrum (a + b (II) looks to some extent different. Also, when we examined this mixture against other mixtures (a+c, a+d and a+e), the changes should be considered the most intense. However, the observed differences are due to properties of “b” that is lactose.

  1. What are the criteria (value range) to qualify the stressed condition as “strong, medium, weak or no impact”?

The criteria used to qualify the stressed condition were as follows: strong – the greatest effect, medium – the intermediate effect, weak – the lowest effect, and no impact – the stressed conditions have not effect on the samples examined. This is due to the fact that the use of the multivariate statistical methods to solution the different problems leads to generalization of outcomes of PCA and HCH calculations. As this generalization is based on the dimensionality reduction of huge data sets or tendency of samples to create the groups of samples with similar characteristics, relationship between the samples examined cannot be given in exact numeric values. Thus, the relationships between the samples can only be referred as a strong, medium or week impact of stressed conditions on their stability. Similar terminology is commonly accepted in the spectroscopy (FT-IR, NIR, Raman, and others), the absorption bands in the spectra are referred to as strong, medium, weak, broad and sharp due to their different intensity.

  1. Please add the lettering scheme for the compounds (a(I) through a+e(II) to the column “Compound” in Table 1 and “sample” in Table 2. This allows for a faster comparison of the table entries to the Figures.

According to the Reviewer’s suggestion, the lettering scheme for the compounds and mixtures examined was inserted to the first column, both in Table 1 and Table 2.

  1. Figure 13 would benefit from a slight overhaul, e.g. axis labels in increments of e.g. 500 cm-1. Please mention FTIR data in the figure caption. Please show the same data ranges for Figure 13 and Figure 6.

According to the Reviewer’s suggestion, Figure 13 was re-drawing into correct format and the term “FT-IR data” was mentioned in the figure caption. Additionally, the same data ranges were inserted both in Figure 6 and 13. In addition, the Figures 4-9 (FT-IR and NIR spectra) were corrected to show the whole spectral ranges measured in our experiments.

  1. 13 A and C, and B and D look very similar while Table 2 lists the changes between stressed and unstressed as strongest for this mixture. Please explain how this analysis translates into the “strong, medium, weak or no impact” categorization.

Figs 13A and 13C, and Figs 13 B and 13 D look very similar but they are not the same. The differences between these figures are compiled in Table 3 (not in Table 2). Table 3 shows the influence of excipients and stressed conditions on the linagliptin, however, this impact is expressed as changes in the linagliptin FT-IR absorption bands. Hence, the influence of excipients and stressed conditions on the linagliptin stability is qualified indirectly, taking into account the changes of FT-IR bands. The greatest changes are expressed as “strong”, the intermediate changes are expressed as “medium”, and so on.

  1. The results for LINA+LAC and LINA+PVP in Table 2 and Table 3 seem to contradict each other. Both show similar behavior in their PC1 and PC2 values, but Table 2 indicates strong vs. no impact in the IR.

The data listed in Table 2 and Table 3 are not the same. Table 2 shows the effect of stressed conditions on linagliptin and its mixtures with excipients, whereas Table 3 shows indirectly the effect of excipients and stressed conditions on the linagliptin stability. Outcomes compiled in Table 2 were developed based on the PCA and HCA calculations using three: DSC, FT-IR and NIR data sets, whereas those compiled in Table 3 were obtained using only PCA based on FT-IR data set. Moreover, the matrices used for PCA calculations were constructed in different way (Materials and Methods), using thermal and spectral data sets for linagliptin, excipients and their mixtures (Table 2), or using only FT-IR data set for linagliptin and one excipients or linagliptin and its mixture with one excipients (Table 3). As PCA and HCA lead to solve the problems using generalization of the data set, the outcomes obtained using different chemometric methods and different matrices construction can be similar, but never the same. This explains slight contradictory in the results obtained.

Reviewer 2 Report

The scientific problem disclosed at the Manuscript entitled "DSC, FT-IR and NIR with chemometric assessment using PCA and HCA for estimation of chemical stability of oral antidiabetic drug linagliptin in the presence of pharmaceutical excipients" refers to pharmaceutical interaction of active ingredients with excipients and is relevant under the dosage form development. A series of modern physicochemical methods were used by authors at the study of stressors influence (high temperature and high humidity) on the stability of tested compounds. 
Authors have conducted a huge scientific work and their results represent great importance for scientific society, however, the spectra inserted into the Manuscript can not be recognized and, as a consequence, evaluated. In order to avoid this, authors should include into the Manuscript just informative pars of spectra followed by labeling the peaks on the curve. 
After this minor revision, I highly recommend the aforementioned Manuscript for publication.   

Author Response

Authors have conducted a huge scientific work and their results represent great importance for scientific society, however, the spectra inserted into the Manuscript can not be recognized and, as a consequence, evaluated.

In order to avoid this, authors should include into the Manuscript just informative pars of spectra followed by labeling the peaks on the curve. 

After this minor revision, I highly recommend the aforementioned Manuscript for publication. 

Thank you for this comment.We are aware that the Figures do not show the perfectly observed changes.Our intention was, above all, to preserve the legibility of the drawings.Because we show 8-9 spectra in one figure, the additional graphics result in a very unreadable image.That is why we decided to simplify the Figures and thoroughly described the changes in the text of our paper.

To illustrate the changes in more detail, in the present version we included individual FT-IR spectra with marked changes as Supplementary Materials.